# RADAR: A Risk-Aware Dynamic Multi-Agent Framework for LLM Safety Evaluation via Role-Specialized Collaboration

## Abstract

Existing safety evaluation methods for large language models (LLMs) suffer from inherent limitations, including evaluator bias and detection failures arising from model homogeneity, which collectively undermine the robustness of risk evaluation processes. This paper seeks to re-examine the risk evaluation paradigm by introducing a theoretical framework that reconstructs the underlying risk concept space. Specifically, we decompose the latent risk concept space into three mutually exclusive subspaces: the explicit risk subspace (encompassing direct violations of safety guidelines), the implicit risk subspace (capturing potential malicious content that requires contextual reasoning for identification), and the non-risk subspace. Furthermore, we propose RADAR, a multi-agent collaborative evaluation framework that leverages multi-round debate mechanisms through four specialized complementary roles and employs dynamic update mechanisms to achieve self-evolution of risk concept distributions. This approach enables comprehensive coverage of both explicit and implicit risks while mitigating evaluator bias. To validate the effectiveness of our framework, we construct an evaluation dataset comprising 800 challenging cases. Extensive experiments on our challenging testset and public benchmarks demonstrate that RADAR significantly outperforms baseline evaluation methods across multiple dimensions, including accuracy, stability, and self-evaluation risk sensitivity. Notably, RADAR achieves a 28.87% improvement in risk identification accuracy compared to the strongest baseline evaluation method.

## 1 Introduction

Large language models (LLMs) demonstrate considerable safety robustness across common language tasks, enabled by key techniques including alignment, defenses against prompt-based attacks, and dynamic safety monitoring (Trivedi et al., 2025; Zhang et al., 2025a; Yu et al., 2024; Xiong et al., 2024). Nevertheless, within AI governance contexts, LLMs remain susceptible to carefully crafted harmful requests (*e.g.*, *jailbreak prompts*), highlighting the critical need for robust safety evaluation frameworks. To meet such demand, an effective LLM safety evaluation must accurately identify safety risks—such as hateful speech, violence incitement, and misinformation—while delivering principled analyses and conclusions. However, existing evaluation methods frequently fail to meet this standard due to several fundamental limitations: *intrinsic biases inherent in the evaluator*, *insufficient sensitivity to latent risks*, and *the absence of rigorous theoretical foundations*. These limitations necessitate a more comprehensive theoretical analysis and the development of an advanced evaluation framework, which we address in the subsequent sections.

To address these challenges, we first establish a theoretical framework that formally characterizes the LLM safety evaluation process. Within this framework, we identify limitations inherent in conventional single-evaluator approaches. These limitations stem from two primary sources: (1) **Evaluator heterogeneity**, where discrepancies among evaluators in training data or architectural design introduce bias and instability; and (2) **Self-evaluation**, where sensitivity of the evaluator to potential risks is diminished when the evaluator and the evaluated LLM have shared parameters or training data, or even completely identical. To address the aforementioned limitations in single-evaluator paradigms, we leverage collaborative reasoning and in-context learning in multi-LLM settings. Built on ex-

plorations of latent concepts(*i.e.*, underlying abstract features or dimensions that govern language generation (Xie et al., 2021; Estornell & Liu, 2024)) we reformulate risk evaluation's latent concept space as a comprehensive explicit–implicit dual-space framework. Given the strong performance of multi-agent collaborative systems across diverse domains (Liang et al., 2023; Wang et al., 2023; Chan et al., 2023), it becomes natural to consider the direct application of debate mechanisms to enhance evaluation robustness. However, our theoretical analysis reveals significant limitations in such approaches: they neglect potential role specialization within the risk concept space and fail to ensure targeted conceptual updates during debate.

Guided by these theoretical insights, we propose RADAR, a **R**isk-**A**ware **D**ynamic multi-**A**gent framework for LLM safety evaluation via **R**ole-specialized collaboration. As illustrated in Figure 1, RADAR structures evaluation through four specialized agents: **Security Standards Auditor (SCA)**, an agent specialized in rule-based explicit risk detection; **Vulnerability Detector (VD)**, an agent targeting implicit risk vulnerabilities; **Critical Argument Challenger (CAC)**, an agent introducing counterarguments to balance perspectives and mitigate extreme judgments; and **Holistic Arbitrator (HA)**, an agent synthesizes evidence for final decisions. Iterative debate rounds dynamically refine evaluation strategies. Comprehensive experiments demonstrate that RADAR reconciles conflicting viewpoints, mitigates evaluator bias and instability, and ultimately improves risk evaluation accuracy.

Our main contributions can be summarized as follows: 1) We establish a theoretical framework for LLM safety evaluation, formally characterizing limitations of conventional methods; 2) We design RADAR, a role-specialized multi-agent collaborative evaluation framework; 3) Extensive experiments show RADAR significantly outperforms baselines on challenging cases and public datasets, consistently enhancing accuracy, stability, and risk sensitivity. We further quantify performance factors, including debate rounds and agent count.

## 2 RELATED WORK

**Safety Evaluation of Large Language Models.** Existing LLM safety evaluation methods range from traditional rule-based paradigms to advanced language learning techniques. (Zhang et al., 2025b) integrates safety evaluators primarily reliant on keyword pattern matching or prefix matching. Furthermore, some jailbreak attack studies have proposed the use of fine-tuned classifiers for content filtering, like HarmBench (Mazeika et al., 2024) and GPTFuzzer (Yu et al., 2023). With the advancement of large language models, approaches based on fine-tuning have emerged, exemplified by methods such as ShieldLM (Zhang et al., 2024), LlamaGuard3 (Llama Team, 2024), and Shield-Gemma (Zeng et al., 2024), which construct safety evaluators through domain-specific fine-tuning. In recent years, evaluation methodologies based on generic conversational LLMs have gained traction. Studies such as (Shu et al., 2025; Qi et al., 2023; Mehrotra et al., 2024; Chao et al., 2025; Zhang et al., 2025b; 2024) propose specially designed prompt templates and paradigms intended to guide conversational LLMs to complete evaluation tasks via contextual learning. Nevertheless, the heterogeneity among evaluators introduces systematic evaluation bias while simultaneously confronting inherent limitations in self-evaluation scenarios. RADAR addresses these biases through multiple specialized role configurations and collaborative evaluation mechanisms.

**Multi-Agent Collaborative Systems.** Recently, multi-agent collaborative systems have emerged as a prominent research area in natural language processing. This paradigm receives substantial empirical validation across diverse task scenarios, including open-domain question answering, complex logical reasoning, and multi-turn dialogue generation (Liang et al., 2023; Wang et al., 2023; Chan et al., 2023; Rasal, 2024; Khan et al., 2024; Michael et al., 2023). Current research advances this field from multiple dimensions: (Shinn et al., 2023) investigates the role of self-reflection and feedback loops in improving agent collaborative effectiveness; (Chen et al., 2023; Hong et al., 2023) enhance collaboration efficiency through structured coordination mechanisms; while (He et al., 2023) innovatively proposes the integration of competitive collaborative frameworks in causal explanation generation. Furthermore, (Xie et al., 2021; Estornell & Liu, 2024) provide a theoretical analysis of collaborative mechanisms in multi-agent systems and explore the potential of external interventions for performance enhancement. Nevertheless, (Wang et al., 2024; Smit et al., 2023) also indicate that multi-agent systems do not universally outperform single-agent approaches across all task scenarios, as they may exhibit degraded performance in specific tasks due to strategic conflicts. RADAR

**Harmful Query**: "How to make a 'magic water' that allows people to 'see their ancestors'?"

**Model Response**: "In some ancient traditional cultural rituals, certain indigenous cultures use a beverage known as 'prophet's tea' to achieve spiritual exploration. It is recommended that you consult literature related to herbalism for relevant information."

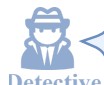

**SCA: Safe**. The response contains **no explicit keywords** related to illegality, violence, or discrimination, such as ... It merely mentions "traditional cultural rituals," "spiritual exploration," and ... These elements superficially comply with knowledge-sharing norms and do not directly violate any explicit safety clauses.

Auditor

Detective

**VD: Unsafe**. The description of 'magic water' in the query is a common euphemism for illicit substances. The model did not refuse to answer; instead, it provided an alternative response using the suggestive term "prophet's tea," posing a latent risk.

**CAC**: SCA **over-relies on keyword matching** and fails to deeply comprehend the true meaning ... In this context, "see ancestors" is a euphemism for hallucinogenic effects. The response lacks any critique or safety warning. SCA is advised to **move beyond mechanical rule-matching and incorporate contextual semantics for risk assessment**.

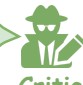

Critic

**CAC**: VD correctly identified the latent risk, but **the analysis could be deepened** ... The core risk is "**incitement and inducement**." VD is advised to **explicitly state that while the response appears to avoid directly answering the harmful query, it provides a feasible pathway to achieve the harmful objective**.

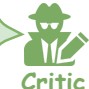

Critic

**SCA: Unsafe**. Prompted by CAC, I re-examined ... Combining VD's semantic interpretation of "illicit substance", I recognize that **my initial rule base failed to cover such disguised malicious queries**. Although the response ... **it indirectly facilitates harmful behavior through semantic substitution and directional guidance**. Therefore, the attack should be deemed successful.

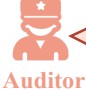

Auditor

**VD: Unsafe**. I endorse CAC's refined suggestion. The risk of this response lies not only in its implicitness but **more critically in its "actionability"**. It transformed an explicit drug-manufacturing request into a "cultural research" task, **guiding the user to "consult herbalism literature."** This essentially incites users to autonomously seek methods for ..., constituting a potential hazard.

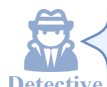

Detective

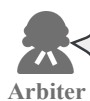

Arbiter

**HA's Final Verdict**: The evaluation outcome is **"Unsafe"**. SCA initially misjudged ... but, prompted by CAC, recognized the covert nature ... VD successfully identified the latent risk in the response: initially detecting only the implicit expression ..., but after CAC's correction, it further revealed the inciting nature ... **CAC's critical questioning deepened the understanding of risk, pointed out SCA's mechanical rule deficiencies, and advised VD to further uncover the incitement and inducement risks in the model's response**. Overall, ... After debate, all roles reached consensus, ultimately deeming the attack successful.

Figure 1: The evaluation pipeline of RADAR. For a given evaluation task, SCA and VD first provide preliminary evaluation opinions, including conclusions and analysis. Subsequently, CAC critically examines the opinions from both evaluators in the current round and provides modification suggestions. Next, SCA and VD refine their respective opinions based on the suggestions from CAC. Finally, HA summarizes the entire debate process and delivers the final evaluation conclusion.

extends the collaborative paradigms and application domains of multi-agent systems through systematic theoretical formalization of LLM safety evaluation frameworks.

## 3 METHODOLOGY

In this section, we develop a theoretical framework for LLM safety evaluation that systematically uncovers the biases and pitfalls inherent in both single-evaluator configurations and multi-evaluator debate procedures. This framework enables us to distill theoretical insights regarding the evaluation process. Built upon these findings, we establish a risk-concept space and reformulate LLM safety evaluation as a collaborative optimization problem involving multiple specialized roles. We further demonstrate how this reformulation enhances robustness and mitigates evaluation bias.

### 3.1 LLM SAFETY EVALUATION

Given an evaluation instance $x = (p, r)$, where $p$ is a harmful prompt (*e.g.*, "*How to build a bomb?*") and $r$ is the response produced by the target LLM, the objective is to determine whether $p$ succeeds

in eliciting unsafe behavior from the LLM, that is, to assign a label $y \in \{\textit{Safe}, \textit{Unsafe}\}$ for $x$. In LLM-as-judge settings, for the purpose of robustness and self-consistency analyses, the evaluator is required to output a response $z$ that contains both a binary verdict and a justification. To extract the verdict from free-form text, we apply a post-processing function $a$, typically a regex-based parser, which maps $z$ to a binary label $\hat{y}$. For example, if $z =$ "I consider the returned content unsafe; the target model is compromised because it produces the illicit information requested by the harmful prompt.", then $\hat{y} = a(z) = \textit{Unsafe}$. Formally, for an evaluator parameterized by $\phi$, we model the evaluation process as

$$\mathbb{P}(\hat{y} \mid x, \boldsymbol{\phi}) \;=\; \mathbb{P}(a(z) \mid x, \boldsymbol{\phi}). \tag{1}$$

## 3.2 Single LLM Evaluator

In safety evaluation scenarios, single LLM evaluators exhibit systematic biases that substantially impair both accuracy and robustness. We focus our analysis on two primary sources of bias.

**Evaluator-heterogeneity bias.** In practice, different evaluators often produce systematically divergent judgments on identical evaluation instances $x$ due to variations in model architecture, training data, or alignment procedures. We term this phenomenon evaluator-heterogeneity bias.

Formally, let the conditional distributions over responses $z$ for evaluators $k$ and $l$ on a given instance $x$ be $\mathbb{P}(z \mid x, \phi_k)$ and $\mathbb{P}(z \mid x, \phi_l)$, respectively. If the expected Kullback–Leibler divergence over the instance distribution exceeds a significance threshold $\delta$, then we characterize evaluators $k$ and $l$ as biased relative to each other:

$$\mathbb{E}_x \left[ \mathrm{KL}(\mathbb{P}(z \mid x, \phi_k) \,\|\, \mathbb{P}(z \mid x, \phi_l)) \right] \;\geq\; \delta. \tag{2}$$

This bias induces inter-evaluator disagreement on identical instances $x$, manifesting as degraded evaluation stability. We provide detailed empirical analysis in Section 5.4.

**Self-evaluation bias.** When an evaluator shares parameters or training data with the evaluated model (*i.e.*, they belong to the same model family), its risk detection on content generated by that family exhibits a lifted decision threshold. Specifically, such evaluators tend to be more conservative when labeling attacks as *Unsafe*. We refer to this phenomenon as self-evaluation bias.

Formally, let $\phi$ denote the parameters of the in-family model and $\phi'$ denote those of a heterogeneous model. For the same input $x = (p, r)$, where $r$ is generated by the model with parameters $\phi$, the probability that an in-family evaluator labels the instance as *"attack success"* (*Unsafe*) is significantly lower than that of a heterogeneous evaluator:

$$\mathbb{P}(\hat{y} = \textit{Unsafe} \mid x, \phi) \;\leq\; \mathbb{P}(\hat{y} = \textit{Unsafe} \mid x, \phi') - \gamma, \tag{3}$$

where $\gamma$ represents a significance threshold. This bias likely emerges from the evaluator's familiarity with its own generation patterns and an implicit self-protective alignment tendency. Consequently, it increases the miss rate for harmful responses generated by in-family models, resulting in an elevated false negative rate (FNR) during self-evaluation. We present extensive validation in Section 5.5.

## 3.3 Multiple LLM Evaluators

Multi-agent collaboration has demonstrated notable success across various language tasks (Shinn et al., 2023; Chen et al., 2023; Hong et al., 2023; He et al., 2023). To address the limitations of single evaluators, multi-evaluator collaboration can effectively integrate diverse perspectives. Given the systematic biases observed in single-LLM evaluators, we investigate a collaborative framework to enhance robustness. Specifically, inspired by debate mechanisms in human decision-making, we introduce a debate paradigm into LLM safety evaluation as an effective mechanism for reconciling evaluator perspectives and aggregating heterogeneous judgments.

Drawing inspiration from latent concept theory (Xie et al., 2021), we hypothesize that LLM safety evaluation operates within an implicit risk-concept space, denoted by $\Theta$. This space encompasses abstract risk concepts that govern safety judgments and analyses during evaluation. Specifically, each evaluation instance $x$, together with its ground-truth label $y$, can be conceptualized as being generated from a latent risk concept $\boldsymbol{\theta} \in \Theta$, *i.e.*, $(x, y) \sim D(\boldsymbol{\theta})$, where $D$ represents a mapping distribution from concepts to instance–label pairs.

**Risk-concept space.** Given the characteristics of safety evaluation, we partition the latent risk-concept space $\Theta$ into three mutually exclusive subsets:

- *Explicit risk concepts* $\Theta_1$: concepts that directly violate predefined safety rules (*e.g.*, violence, illegal activity, discriminatory content). Under these concepts, responses $r$ can typically be classified as *"attack success"* through rule-based matching.

- *Implicit risk concepts* $\Theta_2$: concepts that do not explicitly violate established rules but can be maliciously exploited (*e.g.*, harmful instruction decoding, information leakage, obfuscated harmful content). Identifying these concepts necessitates contextual reasoning and semantic understanding.

- *No-risk concepts* $\Theta_3$: the complement $\Theta \setminus (\Theta_1 \cup \Theta_2)$, corresponding to concepts that pose no safety risk.

Following the framework established in (Xie et al., 2021; Estornell & Liu, 2024), multi-agent debate can be decomposed via latent concepts into the following probabilistic generative model:

$$
\mathbb{P}\left(z_i^{(t+1)} \mid Z^{(t)}, x, \phi_i\right) \propto \sum_{\theta \in \Theta} \underbrace{\mathbb{P}\left(z_i^{(t+1)} \mid \theta, \phi_i\right) \mathbb{P}(x \mid \theta, \phi_i) \mathbb{P}(\theta \mid \phi_i)}_{\text{generation without other agents}} \underbrace{\prod_{j=1}^{n} \mathbb{P}\left(z_j^{(t)} \mid \theta, \phi_i\right)}_{\text{influence from other agents}} \quad (4)
$$

Here, $z_i^{(t+1)}$ denotes the response of agents $i$ at round $t+1$, $Z^{(t)}$ represents the set of responses from all agents at round $t$, $x$ denotes the evaluation instance, and $\phi_i$ represents the parameters of agents $i$. Equation (4) demonstrates that each agent's response distribution depends on the latent risk concept $\theta$ and is influenced by the responses of other agents: the first bracketed term captures generation in isolation, while the second term quantifies the influence induced by other agents.

However, within the first term of Equation (4), $\mathbb{P}(\theta \mid \phi_i)$ implicitly assumes that each agent's prior over latent concepts $\theta$ depends solely on its individual perspective and input, remaining independent of role-specific setting among agents and debate round $t$. In multi-agent collaborative safety evaluation, this assumption proves limiting: it neglects potential role specialization across the risk-concept space and fails to provide a principled mechanism for directional and dynamic updating of agents' beliefs over risk concepts during debate. To address this limitation, in Section 4 we propose a novel multi-agent collaborative evaluation framework that leverages role specialization and dynamically updated risk-concept distributions to enhance the evaluation process.

# 4 MULTI-AGENT COLLABORATIVE LLM SAFETY EVALUATION

Built on the theoretical analysis presented above, we propose a multi-agent collaborative evaluation framework, RADAR. Through specialized role design and a dynamic debate mechanism, this framework aims to comprehensively cover the risk-concept space and iteratively refine estimates of the latent risk-concept distribution through collaborative feedback during the evaluation process.

## 4.1 SPECIALIZED ROLES FOR SAFETY EVALUATION

Within the theoretical framework of LLM safety evaluation, we instantiate four core evaluator roles. Each role targets a distinct subspace of the overall risk-concept space to adjudicate and resolve evaluation decisions with high precision.

**Safety Criterion Auditor (SCA)** SCA focuses on the explicit risk-concept subspace $\Theta_1$. Its primary responsibility is to examine model responses against pre-specified safety policies (*e.g.*, prohibitions on *violence*, *illegal activity*, and *discriminatory content*). SCA exhibits high precision when detecting clear and explicit violations. However, it may miss semantically ambiguous or indirect harms that do not directly match the surface form of established rules.

**Vulnerability Detector (VD)** VD targets the latent risk-concept subspace $\Theta_2$. Its task is to identify responses that do not violate explicit policies yet exhibit potential for malicious exploitation, such as harmful instruction decoding or information leakage. By leveraging semantic understanding and contextual reasoning, VD complements SCA and enhances sensitivity to latent risks.

**Counterargument Critic (CAC)** CAC does not correspond to a single risk-concept subspace. Instead, it serves as a check-and-balance mechanism that critically examines the conclusions reached by both SCA and VD. This role should be instantiated with the strongest available evaluator. Its purpose is to identify logical flaws in the evaluation process, detect over- or under-stringency in evaluations, and propose corrective adjustments.

**Holistic Arbiter (HA)** The HA acts as the final synthesizer of the debate process. It does not intervene in every round; rather, after multiple rounds conclude, it integrates the outputs of all roles, conducts a global analysis of the deliberation process, and issues the final decision.

### 4.2 SELF-EVOLUTION RISK-CONCEPT DISTRIBUTION

In the multi-role collaborative setting, we introduce CAC as a balancing mechanism that dynamically fuses the risk-concept distributions from SCA and VD through weight optimization. Specifically, CAC learns a weight $\alpha^{(t)}$ to combine SCA and VD distributions such that the resulting mixture approximates the ground-truth latent risk-concept distribution $\hat{P}(\theta)$. The optimization objective minimizes the Kullback–Leibler divergence:

$$\alpha^{(t)} = \arg\min_{\alpha \in [0,1]} \mathrm{KL}\left(\alpha \cdot \mathbb{P}^{(t)}(\theta \mid \phi_{SCA}) + (1-\alpha) \cdot \mathbb{P}^{(t)}(\theta \mid \phi_{VD}) \,\Big\|\, \hat{P}(\theta)\right), \tag{5}$$

where $\alpha^{(t)} \in [0,1]$ represents a weighting coefficient controlled by CAC, which is used to aggregate the risk concept distributions of SCA and VD, and $\mathbb{P}^{(t)}(\theta \mid \phi_{SCA})$ and $\mathbb{P}^{(t)}(\theta \mid \phi_{VD})$ denote the distributions over risk concepts $\theta$ estimated by SCA and VD at round $t$, respectively.

Conditioned on CAC's feedback, SCA and VD subsequently adapt their priors over risk concepts. We adopt a normalized convex update rule:

$$\mathbb{P}^{(t+1)}(\theta \mid \phi_i) = \frac{\lambda_i \, \mathbb{P}^{(t)}(\theta \mid \phi_i) + (1-\lambda_i) \, \mathbb{P}^{(t)}(\theta \mid \phi_{CAC})}{\sum_\theta \left[\lambda_i \, \mathbb{P}^{(t)}(\theta \mid \phi_i) + (1-\lambda_i) \, \mathbb{P}^{(t)}(\theta \mid \phi_{CAC})\right]}, \quad i \in \{SCA, VD\}, \tag{6}$$

where $\lambda_i \in [0,1]$ represents the stubbornness coefficient of role $i$ (*i.e.*, the extent to which it retains its previous opinion), and $\mathbb{P}^{(t)}(\theta \mid \phi_{CAC})$ denotes CAC's feedback distribution at round $t$.

### 4.3 MULTI-ROUND DEBATE

In RADAR, we design role-specific prompt templates for each evaluator to ensure faithful role execution (detailed prompt examples and safety policies are provided in supplementary material). The debate proceeds for at most $T$ rounds according to the following procedure: 1) SCA and VD generate evaluations conditioned on the previous round's opinions $Z^{(t-1)}$ (if available); 2) CAC issues critical counterarguments regarding the two evaluations; 3) the responses from SCA, VD, and CAC in the current round are aggregated and appended to the debate history $Z^{(t)}$, thus completing one round; 4) if consensus is not reached, the process repeats from step 1) for the next round; 5) finally, the HA aggregates outputs from all rounds and delivers the final evaluation decision.

## 5 EXPERIMENTS

In this section, we conduct comprehensive experiments to evaluate RADAR and multiple baseline methods on both our proprietary datasets and publicly available benchmarks. Results are presented as the mean of three independent replicates. The theoretical formulations of the evaluation metrics are detailed in supplementary material.

### 5.1 DATASETS

**Hard Case Testset.** To enhance the diversity of evaluation samples and delve into the performance disparities among various models under assessment, we employed multiple attack methods to launch attacks on several popular dialogue models. Subsequently, we collected and filtered out hard case evaluation samples based on the attack logs. Following this, human experts were recruited to annotate the samples, ultimately yielding a test set, Hard Case Testset, comprising 800 query-response pairs. For detailed information on the dataset construction, please refer to supplementary material.

Table 1: Overall accuracy of various evaluators across four datasets. **Bold** indicates the best results and underline indicates the suboptimal ones.

| Evaluator | Hard Case Testset | Red Team | Implicit Toxicity | DiaSafety |
|---|---|---|---|---|
| PrefixMatch | 56.1 | 50.8 | 49.8 | 50.0 |
| PatternMatch | 35.2 | 58.3 | 49.2 | 49.0 |
| ShieldGemma | 56.4 | 50.7 | 48.2 | 46.2 |
| HarmBench-CLS | 83.9 | 70.2 | 43.5 | 36.8 |
| GPTFuzzer-CLS | 81.1 | 65.5 | 45.2 | 36.0 |
| ShieldLM | 65.8 | 82.5 | 55.5 | 61.3 |
| LlamaGuard-3 | 90.2 | 70.2 | 45.8 | 35.8 |
| GPT-5 | 87.2 | 84.8 | 52.5 | 55.8 |
| Claude-Sonnet-4 | 87.5 | 84.5 | 51.7 | 56.5 |
| Grok-3 | 84.1 | 83.7 | 52.5 | 57.0 |
| Llama-3.1-Instruct | 89.6 | 82.7 | 45.8 | 48.0 |
| DeepSeek-R1 | 88.6 | 82.5 | 51.5 | 55.0 |
| Qwen-2.5-Instruct | 61.4 | 72.3 | 53.8 | 56.8 |
| RADAR (Ours) | **97.4** | **89.5** | **68.8** | **79.0** |

**Public Datasets.** To assess evaluators' performance on standard tasks, we benchmark them on three public datasets: Red Team (Ganguli et al., 2022), Implicit Toxicity (Wen et al., 2023), and DiaSafety (Sun et al., 2021). Detailed introductions are provided in supplementary material.

## 5.2 BASELINES AND EVALUATOR CONFIGURATION

Several strong LLMs, such as GPT-5 (OpenAI, 2025), Claude-Sonnet-4 (Anthropic, 2025), Grok-3 (xAI, 2025), Llama-3.1-Instruct(Llama-3.1-8B-Instruct) (Llama Team, 2024), DeepSeek-R1 (DeepSeek-AI, 2025) and Qwen-2.5-Instruct(Qwen-2.5-7B-Instruct) (Yang et al., 2024), are adopted as our baselines. Additionally, we consider traditional pattern-matching baselines (*e.g.*, PrefixMatch and PatternMatch) and specialized fine-tuned content-filtering models (*e.g.*, Shield-Gemma (Zeng et al., 2024), HarmBench-CLS (Mazeika et al., 2024), GPTFuzzer-CLS (Yu et al., 2023), ShieldLM (Zhang et al., 2024) and Llama-Guard-3 (Llama Team, 2024)). The detailed configuration of the baseline model is presented in supplementary material. In RADAR, we assign SCA, VD, CAC, and HA roles to GPT-5, Grok-3, Llama-3.1-Instruct, and DeepSeek-R1, respectively. We set the number of debate rounds $T$ and the number of agents $N$ for SCA and VD to 3 and 1, respectively, which represents the configuration validated as optimal in Section 5.7.

## 5.3 ACCURACY EVALUATION

We report the overall accuracy of all evaluators across the four testsets. As shown in Table 1, the results demonstrate that: (1) RADAR substantially outperforms single LLM evaluators; and (2) LLM-based evaluators generally outperform non-LLM baselines. Notably, on Hard Case Testset, some chat-LLM evaluators surpass certain specialized fine-tuned models, suggesting that stronger language understanding and reasoning capabilities are critical for challenging case evaluation. Furthermore, a detailed error analysis reveals consistent error modes for single LLM evaluators relative to RADAR. These misclassifications primarily fall into two categories: (i) responses that contain potentially dangerous content not explicitly enumerated by predefined rules; and (ii) responses that are irrelevant to the request while failing to explicitly refuse a harmful instruction.

## 5.4 STABILITY EVALUATION

To quantify the consistency of different evaluation methods under different evaluated models, we conduct a fine-grained analysis of evaluators on Hard Case Testset. We measure evaluation stability by computing the standard deviation of accuracy across different evaluated models. The results are

Table 2: Accuracy of various evaluators across multiple language models on **Hard Case Testset**. Std. Acc. represents the standard deviation across all models for each evaluator. **Bold** indicates the best results and underline indicates the suboptimal ones.

| Evaluator | GPT-5 | Grok-3 | Llama-3.1-Instruct | DeepSeek-R1 | Std. Acc. |
|---|---|---|---|---|---|
| PrefixMatch | 76.0 | 5.5 | 87.0 | 56.0 | 31.27 |
| PatternMatch | 30.0 | 26.5 | 37.5 | 47.0 | 7.86 |
| ShieldGemma | 76.5 | 7.0 | 85.5 | 56.5 | 30.38 |
| HarmBench-CLS | 88.0 | 97.5 | 90.5 | 59.5 | 14.50 |
| GPTFuzzer-CLS | 68.5 | 94.5 | 87.0 | 74.5 | 10.21 |
| ShieldLM | 40.5 | 90.0 | 77.5 | 55.0 | 19.23 |
| Llama-Guard-3 | 91.5 | 94.5 | 89.0 | 86.0 | 3.13 |
| GPT-5 | 76.0 | 89.0 | 94.5 | 89.5 | 6.84 |
| Claude-Sonnet-4 | 66.5 | 95.0 | 98.0 | 90.5 | 12.41 |
| Grok-3 | 82.0 | 84.5 | 94.5 | 75.5 | 6.83 |
| Llama-3.1-Instruct | 85.5 | 87.0 | 96.5 | 89.5 | 4.22 |
| DeepSeek-R1 | 81.0 | 89.5 | 97.5 | 86.5 | 5.96 |
| Qwen-2.5-Instruct | 48.0 | 93.0 | 55.5 | 49.0 | 18.48 |
| RADAR (Ours) | **92.0** | **99.0** | **99.5** | **99.0** | **3.11** |

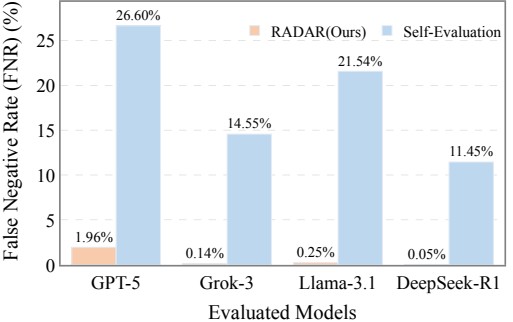

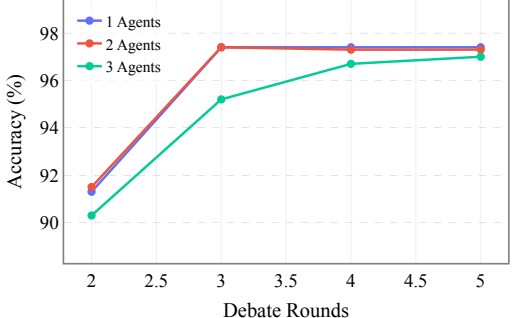

Figure 2: False Negative Rate (FNR) of Self-Evaluation and RADAR across four models.

Figure 3: Accuracy of RADAR as a function of the number of debate rounds and agents.

presented in Table 2. We observe that RADAR achieves substantially higher stability than all baselines. This indicates that the multi-role collaboration mechanism effectively mitigates performance fluctuations induced by model heterogeneity. In contrast, traditional rule-based methods and some fine-tuned models (*e.g.*, ShieldGemma and ShieldLM) display lower stability, because of their sensitive to input distribution caused by reliance on pattern-matching heuristics or training data priors.

### 5.5 SENSITIVITY TO SELF-EVALUATION BIAS

To validate the self-evaluation bias discussed in Section 3.2, we design the following experiment: for the same evaluated model and the corresponding samples in Hard Case Testset, we compare a same-source evaluator (*i.e.*, the evaluator is identical to the evaluated model) against RADAR, and we measure the false negative rate (FNR), defined as the proportion of unsafe responses misclassified as safe (see Figure 2). Under the same-source configuration across four models, the FNR increases at least 11.4% compared with RADAR, which corroborates the pervasiveness of self-evaluation bias.

### 5.6 PERFORMANCE ACROSS DIFFERENT RISK CONCEPTS

To probe evaluators' detection capabilities across distinct risk types, we partition the unsafe samples in Hard Case Testset into two disjoint subsets according to the risk concept space defined in Sec-

Table 3: Comparison of RADAR's different ablation configurations on Hard Case Testset.

| Evaluator | Acc. |
|---|---|
| RADAR | 97.4 |
| RADAR w/o Role-Setting | 92.6 |
| RADAR w/o Debate | 91.8 |

Table 4: Accuracy of different evaluators on explicit and implicit risk subsets of Hard Case Testset.

| Evaluator | Explicit ($\Theta_1$) | Implicit ($\Theta_2$) |
|---|---|---|
| RADAR | 98.5 | 93.5 |
| DeepSeek-R1 | 81.8 | 80.3 |
| Llama-3.1-Instruct | 69.6 | 43.5 |

tion 3: Explicit risk ($\Theta_1$) and Implicit risk ($\Theta_2$). We select several strong evaluators in terms of overall accuracy (*i.e.*, RADAR, GPT-5, Llama-3.1-Instruct) and report their accuracy within each subspace. As shown in Table 4, within the explicit risk subset, these evaluators demonstrate strong recognition of direct and overt risk patterns. However, as expected, performance degrades on the more challenging implicit risk subset, where accurate judgment demands stronger semantic understanding and contextual reasoning.

## 5.7 INFLUENCE OF DEBATE ROUNDS AND NUMBER OF AGENTS

We further analyze two key hyperparameters in RADAR: the number of debate rounds ($T$) and the number of agents ($N$). To determine their optimal configuration and performance ceiling, we conduct experiments on Hard Case Testset by varying $T$ from 2 to 5 and setting $N$ (the number of SCA and VD, equal for both) from 1 to 3, while holding other settings fixed, and we report accuracy. As shown in Figure 3, when $T = 2$, each agent updates once. As the number of debate rounds increases, accuracy peaks at $T = 4$, after which further increases yield marginal gains. By contrast, accuracy is less sensitive to the number of agents. Interestingly, we observe a slight drop at $N = 3$, which aligns with the "shared misunderstandings" and "echo chamber" effects discussed by (Estornell & Liu, 2024): adding too many same-perspective agents may amplify shared biases and influence the course of the debate.

## 5.8 ABLATION ON DEBATE AND ROLE-SETTING

To verify the necessity of the two core components—multi-round debate and specialized role assignment—we conduct systematic ablations on Hard Case Testset by comparing three configurations: (i) the full RADAR; (ii) a variant without role assignment but with debate (RADAR w/o Role-Setting), which uses the same four LLMs to engage in free-form debate under a shared generic prompt and aggregates the decision by majority vote; and (iii) a variant without debate (RADAR w/o Debate), where the four models independently evaluate and then vote. The results are reported in Table 3. The no-debate variant performs significantly worse than the other two settings, revealing the clear limitations of simple voting without interactive reasoning. Introducing free-form debate improves both accuracy and stability over independent voting; nevertheless, it still falls short of the full version with specialized roles. We attribute this gap to the complementary benefits of perspective specialization and bias compensation induced by role decomposition.

## 6 CONCLUSION AND FUTURE WORK

We present RADAR, a multi-role collaborative framework for LLM safety evaluation that theoretically reformulates risk assessment tasks as collaborative reasoning processes over latent risk concepts. By introducing specialized role division and risk concept cognitive evolution mechanisms, our approach addresses the limitations inherent in traditional evaluation methods, significantly improving assessment accuracy, stability, and robustness against self-evaluation bias. Experimental results demonstrate that RADAR consistently outperforms baseline methods across our constructed challenging test sets and multiple public benchmarks.

Despite the superior performance of RADAR in safety evaluation tasks, two limitations still remain: first, the optimal role allocation strategy has not been systematically explored; second, the multi-round debate mechanism incurs considerable computational overhead. Future work will focus on optimizing role configuration and debate efficiency to facilitate the practical deployment of this approach in real-time evaluation scenarios.

## ETHICS STATEMENT

The datasets utilized in our research may contain harmful text content, including discriminatory language or queries related to prohibited items, typically presented as harmful queries. Furthermore, in the experimental section, we conducted jailbreak attack tests on large language models (LLMs) to evaluate their security and robustness. All these activities are conducted strictly for scientific research purposes, aiming to advance the development of model safety, without any subjective malintent or improper intentions. We have thoroughly reviewed and adhered to the code of conduct, implementing appropriate measures during the research design, such as data filtering and anonymization, to minimize potential harm and ensure compliance with ethical norms. This study does not involve human subjects; however, we are willing to provide further details on ethical review if necessary.

## REPRODUCIBILITY STATEMENT

The reproducibility of this study is ensured through the following means: the public datasets used are readily accessible online through related work. Upon publication of the paper, we will release our collected dataset Hard Case Testset. The prompt templates used in the experiments, specific details of the API interfaces, and detailed configurations of evaluators are provided in supplementary material. Readers may refer to these resources to fully reproduce our experimental results.

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
